# Protoplasmic Perivascular Astrocytes Play a Crucial Role in the Development of Enlarged Perivascular Spaces in Obesity, Metabolic Syndrome, and Type 2 Diabetes Mellitus

**Melvin R. Hayden** [ID]

Department of Internal Medicine, Endocrinology Diabetes and Metabolism, Diabetes and Cardiovascular Disease Center, University of Missouri School of Medicine, One Hospital Drive, Columbia, MO 65211, USA; mrh29pete@gmail.com; Tel.: +1-573-346-3019

**Abstract:** Astrocytes (ACs) are the most abundant cells in the brain and, importantly, are the master connecting and communicating cells that provide structural and functional support for brain cells at all levels of organization. Further, they are recognized as the guardians and housekeepers of the brain. Protoplasmic perivascular astrocyte endfeet and their basal lamina form the delimiting outermost barrier (glia limitans) of the perivascular spaces in postcapillary venules and are important for the clearance of metabolic waste. They comprise the glymphatic system, which is critically dependent on proper waste removal by the pvACef polarized aquaporin-4 water channels. Also, the protoplasmic perisynaptic astrocyte endfeet (psACef) are important in cradling the neuronal synapses that serve to maintain homeostasis and serve a functional and supportive role in synaptic transmission. Enlarged perivascular spaces (EPVS) are emerging as important aberrant findings on magnetic resonance imaging (MRI), and are associated with white matter hyperintensities, lacunes, and aging, and are accepted as biomarkers for cerebral small vessel disease, increased obesity, metabolic syndrome, and type 2 diabetes. Knowledge is exponentially expanding regarding EPVS along with the glymphatic system, since EPVS are closely associated with impaired glymphatic function and waste removal from the brain to the cerebrospinal fluid and systemic circulation. This review intends to focus on how the pvACef play a crucial role in the development of EPVS.

**Keywords:** blood–brain barrier; endothelial glycocalyx; enlarged perivascular spaces; metabolic syndrome; neuroinflammation; neurovascular unit; obesity; perivascular spaces; perivascular unit; T2DM



## 1. Introduction

Perivascular spaces (PVS) are fluid filled spaces that ensheathe pia vessels as they dive into the cortical grey and white matter of the central nervous system (CNS). The pia arteries and precapillary arterioles PVS are known to deliver cerebrospinal fluid (CSF) to the interstitium, while the postcapillary venules and veins are known to deliver primarily interstitial fluid (ISF), metabolic waste (MW), and some residual admixed CSF to the subarachnoid space (SAS) for eventual disposal from the brain to the systemic circulation (Figure 1) [1–4].

Protoplasmic perivascular astrocyte endfeet (pvACef) adhere tightly to the basement membrane (BM) of the neurovascular unit (NVU) shared by both the brain endothelial cell(s) (BECs) and pericyte(s) (Pcs) via their pvACef basal lamina, also termed the glia limitans (GL). PvACef are responsible for integrating the vascular mural cells (BECs and Pcs) of the NVU to nearby regional neurons [1–4]. PvACef allow for NVU coupling, which is fundamental for the regulation of regional capillary cerebral blood flow (CBF) by both astrocyte and neuron-derived chemical messengers that provide for functional hyperemia that is known as neurovascular coupling [1,5,6]. PvACef are surrounded by the neuropil, which is comprised primarily of dendritic synapses and unmyelinated

neurons—interneurons with traversing myelinated neurons and an extracellular matrix (ECM) interstitial space (ISS) between these cellular structures.

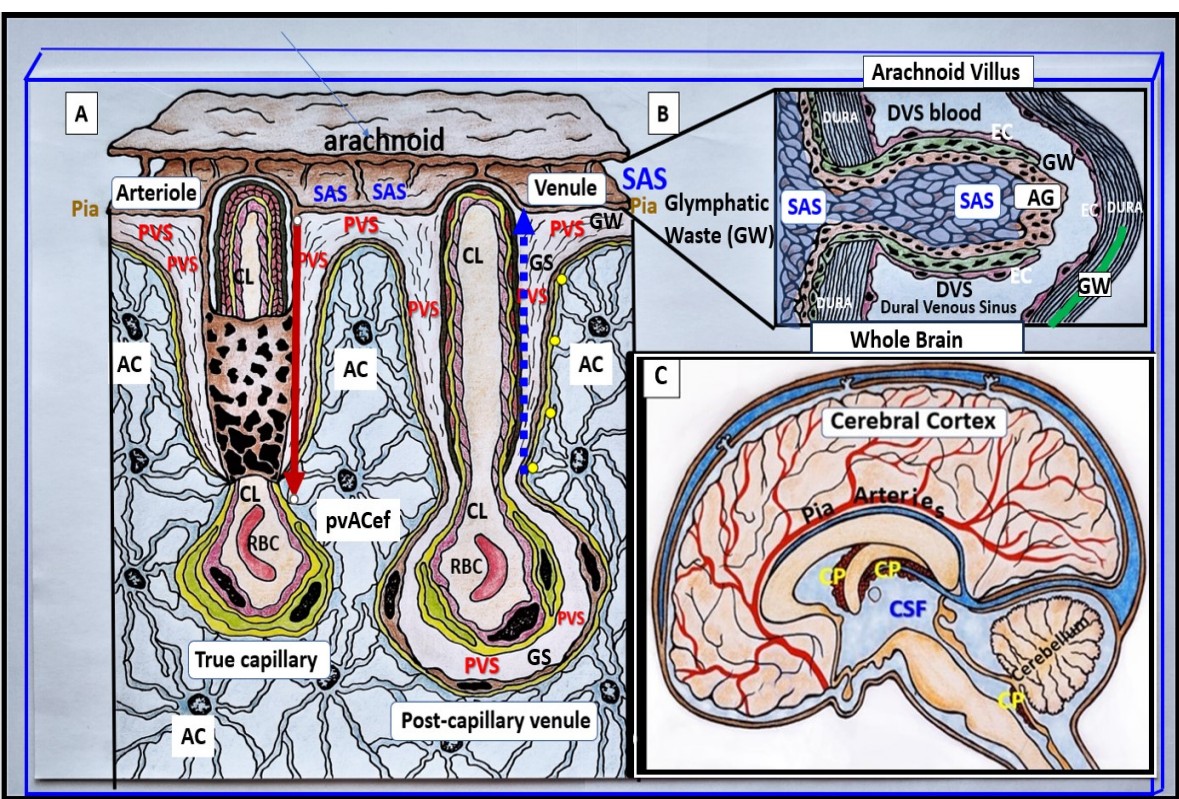

**Figure 1.** Perivascular spaces (PVS) ensheathe the pia arteries and veins, and deliver cerebrospinal fluid (CSF) to the brain and remove interstitial fluid (ISF) and metabolic waste (MW) from the brain for disposal in the CSF. (**A**) illustrates the arteriovenous capillary system and each of their ensheathing PVS with the red arrow depicting incoming CSF and the blue dashed arrow depicting the efflux of the ISF and MW via the PVS to the subarachnoid space (SAS) initially and hence to the CSF for disposal into the systemic circulation via the arachnoid villus (**B**). (**B**) depicts a single arachnoid villus and illustrates the removal of the glymphatic MW to the dura lymph and/or dural venous sinus (DVS) for delivery to the systemic circulation. This mixture (ISF, MW, and CSF) from the PVS—glymphatic system (GS) can then leave the brain through the perivenous space and eventually into the meningeal lymphatics, to cervical lymphatics, and the systemic circulation. (**C**) demonstrates the CSF that provides structural support and bouncy protection, nourishment, and waste removal. Thus, the CSF is necessary for maintaining homeostasis of the ISF and creating a stable environment for the brain parenchyma that is imperative for maintaining normal homeostasis and neuronal function. Image provided with permission from 4.0 [2]. AC = astrocyte; pvACef = protoplasmic perivascular astrocyte endfeet; AG = arachnoid granulation(s); CL = capillary lumen; CP = choroid plexus; DVS = dural venous sinus; RBC = red blood cell.

PvACef with their basal lamina form the delimiting outermost nanosized membrane barrier of perivascular spaces (PVS), which is also referred to as the glia limitans (GL), while the innermost barrier is the basement membrane (BM) of the neurovascular unit (NVU) brain endothelial cell(s) (BECs) and pericytes (Pcs) (Figure 2) [2,3,7].

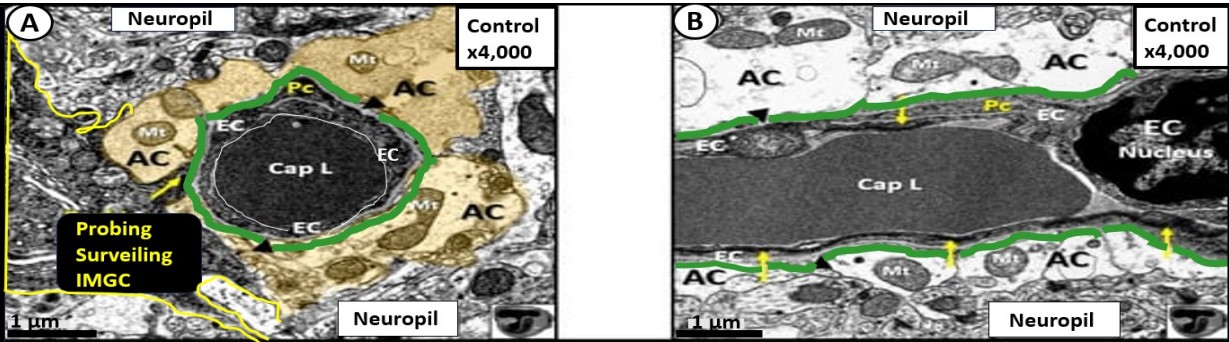

**Figure 2.** Normal ultrastructure neurovascular unit (NVU) morphology in control wild-type non-diabetic models (C57BL/KsJ) to emphasize the perivascular astrocyte endfeet (pvACef). (**A**) demonstrates a cross-section view of the NVU and also depicts an electron-dense ramified interrogating microglia cell (iMGC) (yellow arrow) surveilling the NVU. The NVU capillary consists of brain endothelial cells (BECs) encircling a capillary lumen (Cap L, thin white line) whose basement membrane (BM arrowheads) splits to also include the pericyte (Pc) foot process. Note how the pseudo-colored golden perivascular astrocyte endfeet (AC-pvACef) encompass and tightly abut to the capillary EC and Pc BMs. Importantly, note that the AC clear zone in panel B was pseudo-colored golden to emphasize its importance to the NVU, while it remains as a clear-zone (white) with a reduced electron-dense cytoplasm as compared to other cells within the brain, and represents not only a golden halo, but also a clear zone or corona of pvACef surrounding the BEC and Pc cells and their BMs of the NVU as in (**B**). (**B**) demonstrates a longitudinal view, which illustrates the electron-lucency of the AC clear zone halo or corona (white) that tightly abuts and encircles the NVU BEC and Pc BMs. Note the BEC nucleus (far right side) and the highly electron-dense tight junctions/adherens junctions (TJ/AJ) complex (yellow arrows) that are not readily visible in panel A. Also, note that the mitochondria (Mt) have an electron-dense Mt matrix and that cristae may be noted even at this magnification. Additionally, note that the NVU is encompassed by the glia limitans (green line) of the outermost abluminal pvACef and the neuropil. Magnification ×4000; scale bar = 1 μm. This heavily modified image was provided by CC 4.0 [7]. Cyan green line = perivascular astrocyte endfeet basement membrane-lamina or glia limitans perivascularis.

The glia limitans (GL) also consists of the pvACef basal lamina—BM in the peri-meningeal barrier that is known as *glia limitans superficialis* or *externa*, whereas this barrier surrounding the NVU is defined as *glia limitans perivascularis and further, any* substance entering the central nervous system (CNS) from the blood or cerebrospinal fluid (CSF) must cross the GL [8].

There are three basic types of astrocytes (ACs) that consist of (1) protoplasmic ACs found primarily in the grey matter cortex and are responsible for pvACef and perisynaptic astrocyte endfeet (psACef); (2) fibrous ACs found primarily in the white matter that are important for myelin maintenance and remyelination with interaction among oligodendrocytes and oligodendrocyte precursor cells; (3) peripheral astroglial processes (PAPs) ACs that are responsible for AC cytoplasmic extensions to the pvACef of the NVU and psACef that are known to cradle the synapses [8,9].

ACs are the most abundant cells in the brain and, importantly, are the master connecting, communicating, continuing, and creating cells (in the case of the creation of the perivascular unit (PVU) and its normal PVS and pathologic enlarged perivascular spaces (EPVS) in the postcapillary venule) of the brain. The ACs connect with the NVU via pvACef, and synapses via the perisynaptic astrocyte endfeet (psACef); the fibrous ACs connect to the myelinated neurons in the white matter, and connect to communicate with one another to create the AC syncytium via gap junction connexins (Figure 3) [7–13].

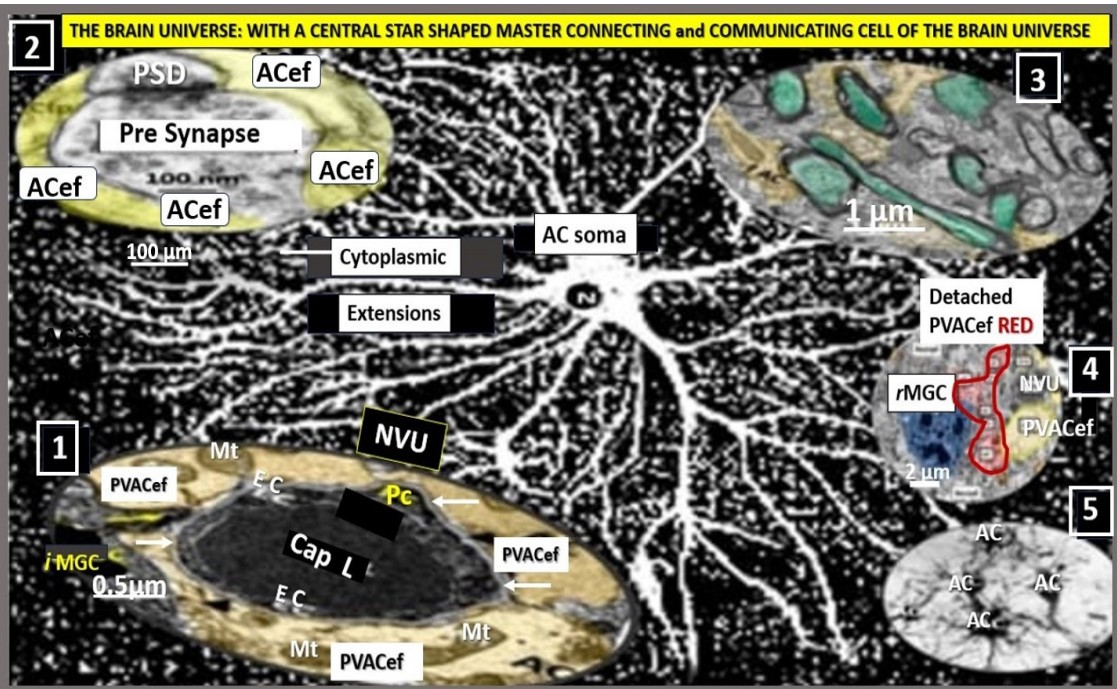

**Figure 3.** Astrocyte(s)' (ACs) connection/communication with other astrocytes and multiple cells within the brain universe. ACs are ectodermal, neuroepithelial-derived, and star-shaped cells that provide protection, defense, and homeostasis of the brain and spinal cord. This montage illustration of hand-drawn images and transmission electron micrographs allows one to observe the resident non-reactive protoplasmic AC soma, as seen in the center of this image with multiple cytoplasmic extensions, that represents a single peripheral astroglial process (PAP) AC that connects to virtually every cell within the brain universe. ACs may thus be considered the master connecting cells of the central nervous system (CNS) against a background of distant white dots representing even more ACs stars within the brain universe from control C57BL/6J models (hand-drawn computer-assisted illustration from toluidine blue stained images). Importantly, the AC-to-AC connections via gap junction connexins such as Cx43 allow them to form a syncytium to coordinate brain function and proper cognition (insert number 5). Insert 1 depicts the protoplasmic perivascular astrocyte endfeet (pvACef) processes (pseudo-colored golden) connecting to the neurovascular unit (NVU) capillary, which allows for neurovascular coupling in frontal cortex layer III in a non-diabetic control female C57BL/6J mouse model at 20 weeks of age. Insert 2 illustrates the protoplasmic perisynaptic astrocyte endfeet (psACef) cradling of pre- and post-synaptic neurons and emphasizes its importance to the tripartite cradle synapse in cortical layer III. Insert 3 illustrates the connection of the fibrous ACs (pseudo-colored yellow-gold and to the formation of myelin) to myelinated axons with axoplasm (pseudo-colored cyan) of neurons in white matter in a control at 20 weeks C57BL/6J. Insert 4 illustrates the lost connections between a reactive microglial cell (*r*MGC) (pseudo-colored blue) and multiple reactive detached and separated ACs (pseudo-colored red) adjacent to a neurovascular unit (NVU) with a single intact non-reactive ACs (pseudo-colored yellow) in diabetic *db/db* model cortical layer III at 20 weeks of age. Insert 5 demonstrates AC-to-AC connections in cortical layer III in control models (hand-drawn computer-assisted illustration of light microscopic toluidine blue stained images from control C57BL/6J models) via gap junction connexins. Only Inserts 1–4 have scale bars of 0.5 μm, 100 nm, 1 μm, and 2 μm, respectively. The background is also a hand-drawn computer-assisted image derived from control C57BL/6J models toluidine blue stained models and does not have a scale bar. This highly modified image is provided with permission from CC 4.0 [13]. ACfp = protoplasmic astrocyte endfeet; ACPVef = astrocyte perivascular endfeet; Cap L = capillary lumen; EC = brain endothelial cell; iMGC = interrogating microglial cell; Mt = mitochondria; N = nucleus; Pc = pericyte; PSD = post-synaptic density; PVACef = perivascular astrocyte endfeet; rMGC = reactive microglia cell; psACef = perisynaptic astrocyte endfeet.

ACs are capable of enacting most housekeeping and guardian homeostatic functions in the brain, from structural support to controlling molecular homeostasis and regulation of CBF, synaptogenesis, neurogenesis, and additionally development of the nervous system [11]. A brief summary of the homeostatic functions of ACs (via pvACef and perisynaptic ACef) include molecular homeostasis, which includes ion homeostasis (of potassium, chloride, and potassium), regulation of pH, water transport and homeostasis via aquaporin-4 (AQP4), and neurotransmitter homeostasis (including glutamate, gamma-aminobutyric acid (GABA), adenosine, and monoamines); systemic homeostasis, including chemosensing ($O_2$, $CO_2$, pH, Na+, and glucose), regulation of energy balance and food intake, and sleep homeostasis; cellular and network homeostasis, including neurogenesis, neuronal guidance, synaptogenesis, synaptic maintenance, elimination, and plasticity; metabolic homeostasis, including NVU formation and maintenance, support of NVU, CBF, metabolic support and maintenance, and glycogen synthesis and storage; organ homeostasis, including the control of the NVU BBB, and the lymphatic and glymphatic systems as partially represented in Figures 2 and 3 [8–11]. Additionally, ACs act as a major supplier of energy via glycogen storage and glycolysis, as well as supplying antioxidant reserves such as glutathione (GSH) and superoxide dismutase (SOD), and growth factors such as brain-derived growth factor transforming growth factor-β (TGFβ). ACs also define many aspects of synapse formation, plasticity, protective function, synaptic maintenance, and elimination [11,12]. It is very important to note that human studies may not always conform to the findings of rodent models because pvACs in the neocortex are much larger in diameter (2.6-fold), have longer extending cellular extensions (10-fold), and have greater complexity and diversity than in rodent models [11,14].

The large AC cellular presence in the brain and their vast cell–cell communication via gap junction connexins may be viewed as the brain's functional syncytium [8]. The relationships among the pvACef and the NVU (including ECs, Pcs, and their shared outer basement membrane, as well as the cell–matrix attachments via dystroglycans and integrins of the pvACef to NVU BMs) are essential for proper homeostasis and function [7,13,15,16].

This review intends to focus not only on ACs and specifically pvACef of the NVU BBB but also the postcapillary venule perivascular unit (PVU) and its normal PVS and the pathologic transformation to the EPVS that become affected by multiple neurotoxicities. These toxicities include: neuroinflammation, reactive oxygen species (ROS), reactive oxygen, nitrogen, sulfur species (RONSS) and the reactive species interactome (RSI), and the breeching of the outermost abluminal layer of the EPVS GL to allow the passage of proinflammatory leukocytes to enter the neuropil interstitial spaces (ISS) and travel throughout the brain along with ISF flow. Importantly, the PVS within the PVU provide an anatomic conduit for the passage of ISF and MW that is termed the glymphatic space or glymphatic system (GS) [17]. Once this GS becomes dysfunctional, there is an accumulation of neurotoxic waste, and misfolded proteins, neuroinflammation, ROS-RSI, and increased reactive microglia cells (rMGCs). Further, this combination will result in dysfunction and/or damage not only to the pvACef of the NVU but also to the supportive and protective psACef that control synapse formation and function [18] and could detach, retract, undergo aging asthenia, leaving the synapse vulnerable to dysfunction and damage with detachment for the pre- and post-synaptic neuron psACef similar to the detachment of pvACef at the NVU BBB. These series of events could result in a synaptopathology with impaired synaptic transmission and impaired cognition that could eventually result in neurodegeneration over time [19,20].

## 2. Metabolic Disorders: Obesity, Metabolic Syndrome (MetS), Type 2 Diabetes Mellitus (T2DM), and Global Aging

The triad of obesity, MetS, and T2DM plus advanced global age are currently global societal problems that are expected to grow over the coming decades. T2DM of this triad and neurodegenerative diseases (including cerebrocardiovascular disease, cerebral small vessel disease, and thrombotic or hemorrhagic stroke) are anticipated to develop into

aging-related EPVS. Also, since the global population is currently one of the oldest, it is expected to continue to increase in frequency over the next 2–3 decades, such that we will observe these two groups merge and increase in numbers [2,21–23]. Obesity and visceral adipose tissue predispose to EPVS-impaired synaptic transmission and impaired cognition, and neurodegeneration over time [24,25]. Insulin resistance (IR), brain insulin resistance (BIR), and MetS also result in brain remodeling (Figure 4) [26–29].

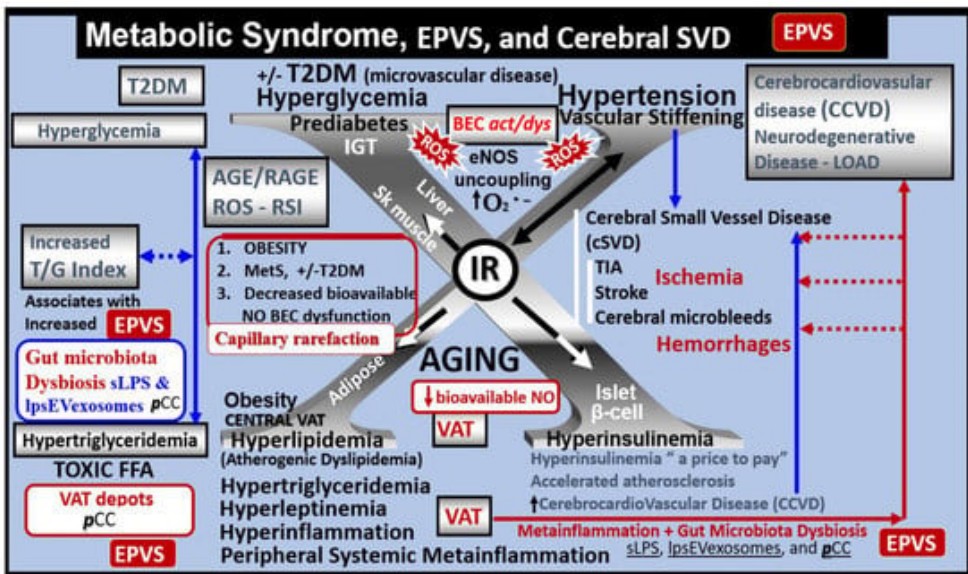

**Figure 4.** Metabolic syndrome (MetS), enlarged perivascular spaces (EPVSs), and cerebral small vessel disease (cSVD). The visceral adipose tissue (VAT), obesity, and hyperlipidemia (atherogenic dyslipidemia) located in the lower left-hand side of the letter X appear to drive the MetS and insulin resistance (IR) central with the other three arms of the letter X, which include the associated hyperinsulinemia to compensate for IR (lower right), hypertension, vascular stiffening (upper right), and hyperglycemia (upper left), with or without manifest type 2 diabetes mellitus (T2DM). Follow the prominent closed red arrows emanating from VAT to cerebrocardiovascular disease (CCVD), SVD, TIA, stroke, cerebral microbleeds, and hemorrhages. Brain endothelial cell activation and dysfunction (BEC*act/dys*), with their proinflammatory and prooxidative properties, result in endothelial nitric oxide synthesis (eNOS) uncoupling with increased superoxide ($O_2^{\bullet-}$) and decreased nitric oxide (NO) bioavailability. Importantly, note that obesity, MetS, T2DM, and decreased bioavailable NO interact to result in capillary rarefaction that may allow EPVS to develop, which are biomarkers for cerebral SVD. Figure markedly adapted with permission from CC 4.0 [2]. AGE = advanced glycation end-products; RAGE = receptor for AGE; AGE/RAGE = advanced glycation end-products and its receptor interaction; βcell = pancreatic islet insulin-producing beta cell; FFA = free fatty acids—unsaturated long chain fatty acids; IGT = impaired glucose tolerance; LOAD = late-onset Alzheimer's disease; ROS = reactive oxygen species; RSI = reactive species interactome; Sk = skeletal: TG Index = triglyceride/glucose index; TIA = transient ischemia attack.

Additionally, T2DM is known to be associated with brain remodeling with cognitive impairment and dysfunction (CID) and EPVS [12,30–39], and it is commonly accepted that age is the strongest risk factor for the development of EPVS, while hypertension, age plus hypertension, and diabetes were still three risk factors for those of 45 years old or under [40].

## 3. Postcapillary Venule Perivascular Unit (PVU), Normal Perivascular Spaces (PVS), and Transformation to Pathological Enlarged Perivascular Spaces (EPVS)

Troili et al. [41] conceived the concept of the PVU that resides adjacent to NVU and contains both the postcapillary venular normal PVS that undergoes the pathologic remodeling change of transformation from normal PVS to the remodeled EPVS (Figure 5) [41,42].

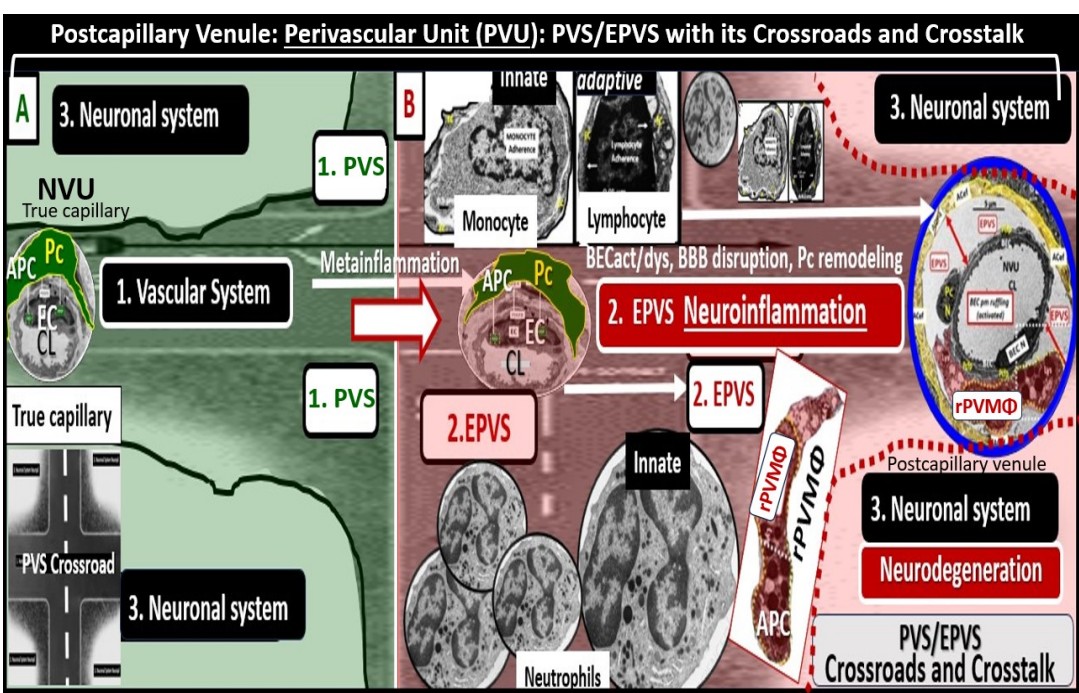

**Figure 5.** The postcapillary venule, the perivascular unit (PVU), and the perivascular spaces (PVS) serve as an anatomical crossroad or intersection for the vascular, neuroinflammatory, and neuronal systems to provide crosstalk communication associated with the metainflammation of obesity, metabolic syndrome (MetS), and type 2 diabetes mellitus (T2DM) in addition to providing the anatomical substrate for the glymphatic system. The PVU is comprised of panels A and B. (**A**) demonstrates the PVU with a normal PVS of the vascular system in control models with the dark green background. Note the highway PVU crossroad icon in the lower left panel. (**B**) depicts the EPVS (2.) with their resident reactive perivascular macrophage (rPVMΦ pseudo-colored red) and leukocytes (neutrophils, monocytes, and lymphocytes) that have undergone diapedesis via primarily paracellular routes via the activated BECs (aBECs), which enter the EPVS and comprise step 1 of the two-step process of leukocytes entering the neuropil interstitial space (ISS). These leukocytes not only undergo cellular crosstalk with the aBECs but also with one another in addition to the resident perivascular macrophage (rPVMΦ), the pericyte (Pc), and the perivascular astrocyte endfeet (pvACef) within the PVU to result in EPVS, impaired cognition, and neurodegeneration. It is important to note that both the Pc and the rPVMΦ are known to be antigen-presenting cell(s) (APCs). Additionally, the reactive leukocytes are capable of generating huge amounts of reactive oxygen species—oxidative stress and the secretion of matrix metalloproteinases (MMP-2, -9) that are capable of degrading the outermost boundary of the EPVS pvACef glia limitans to allow for the second step of leukocyte entry into the neuropil interstitial spaces to result in neuroinflammation and subsequent neurodegeneration. The PVU with its PVS/EPVS are an anatomical crossroad and along with its multiple cellular crosstalk result in a self-perpetration or vicious cycle of brain injury and response to injury wound healing to result in neuroinflammation, impaired cognition, and neurodegeneration. Additionally, it is important to note that the PVS form the anatomical conduit for the glymphatic system to deliver metabolic waste and toxins from the interstitial fluid (ISF) and provide the crosstalk communication necessary for neuroinflammation to develop within the PVS/EPVS of the PVU. The increased neuroinflammation that occurs within the PVS/EPVS of the PVU will develop considerable metabolic waste debris that will slow and cause stagnation of flow with delayed clearance of metabolic waste from the ISF to the subarachnoid space (SAS) and cerebrospinal fluid (CSF) to result in further dilation of the PVS of the PVU. Adapted with permission from CC 4.0 [42]. BBB = blood–brain barrier; BECact/dys = brain endothelial cell activation/dysfunction; CL = capillary lumen; EC = brain endothelial cells; EPVS = enlarged perivascular space; Pc = pericytes; PVS = perivascular space; PVU = perivascular unit; rPVMΦ = resident reactive perivascular macrophage(s).

The cellular composition of the PVU consist of BEC, Pc, interrogating microglia: neurons and their axons that synapse to the pvACef that line the PVU, resident perivascular macrophage(s) (rPVMΦ), and leukocytes that have been taken up into the PVU following activation of the BECs [41]. The PVS within the PVU are of critical importance because they are the anatomic construct that serves as the conduit for the recently discovered glymphatic space [17].

Multiple hypotheses and overlapping mechanisms are thought to be responsible for the enlargement of PVS to result in EPVS, which include at least four hypotheses, as follows: (1) arterial stiffness due to vascular remodeling via arteriolosclerosis and associated spiraling; (2) misfolded protein aggregation such as amyloid beta and tau; (3) brain atrophy and/or loss of myelin; (4) BBB disruption with increased permeability of fluids, plasma proteins, and inflammatory cells, which result in the accumulation of phagocytic debris and obstruction with dilation to develop EPVS [2,41,42]. Also, Yu et al. have proposed that either excessive CSF inflow or a decrease of ISF/CSF outflow or efflux due to neuroinflammatory mechanisms within the PVU as depicted in Figure 5 and listed as number 4 previously, are likely to be responsible for EPVS [2,4].

Recently, Shulyatnikova and the author have recognized and supported microvascular capillary rarefaction as an additional hypothesis for the development of EPVS [2,3,42]. Microvascular capillary rarefaction (loss of capillaries in the brain with decreased capillary density) are frequently found in obesity, MetS, and T2DM [2,3,42,43]. Capillary rarefaction in the brain has recently been found to be associated with an increase in obesity, MetS, and T2DM [44–47]. Capillary loss due to rarefaction leaves an empty space within the confines of the PVUs' PVS that would subsequently fill with ISF, and this would allow for an increase in the percentage of total fluid volume within the PVS (Figure 6) [2,4,43].

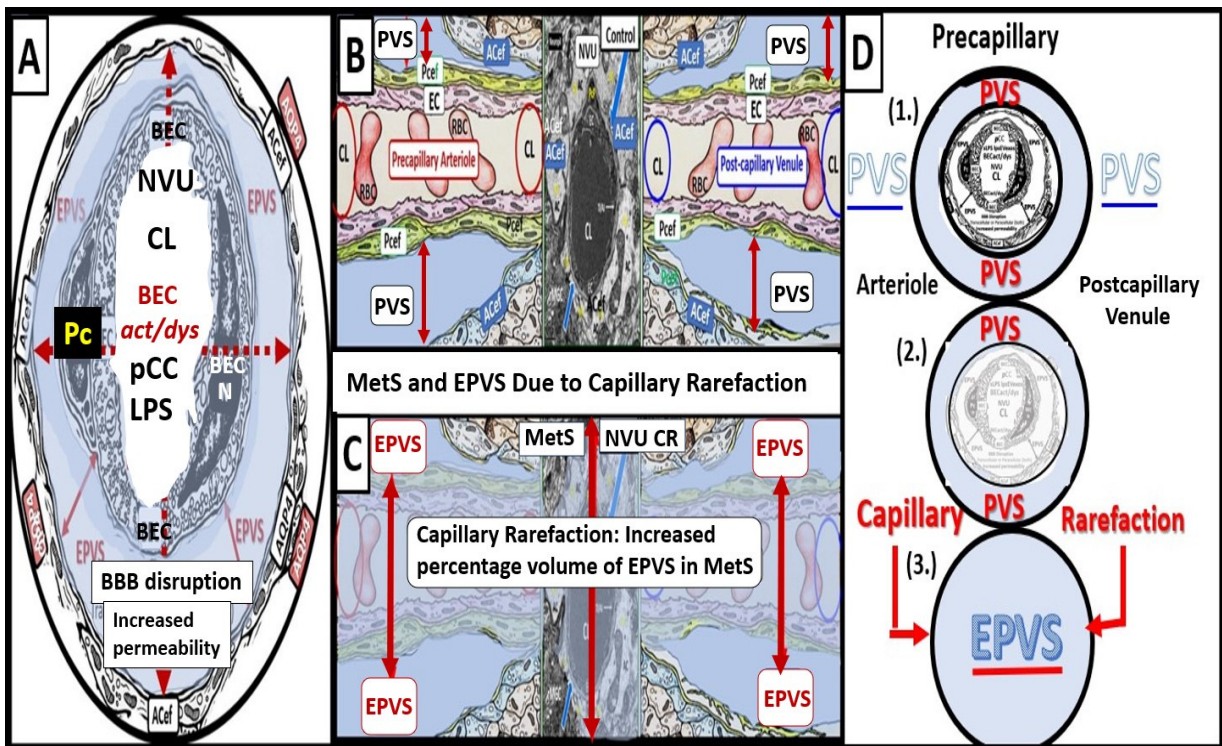

**Figure 6.** Cross and longitudinal sections representative of pre- and postcapillary arterioles and venules with an ensheathing perivascular space (PVS). (**A**) depicts a cross-section of a capillary surrounded by PVS (solid double red arrows and light blue color) and its increase in total volume to become an enlarged perivascular space (EPVS) (dashed double red arrows), which represents capillary rarefaction. Note the AQP4 red bars that are associated with the perivascular astrocyte endfeet. (**B**) demonstrates

a control longitudinal precapillary arteriole, postcapillary venule, and a neurovascular unit (NVU) capillary that runs through an encompassing PVS (light blue). (**C**) depicts capillary rarefaction (CR) in a longitudinal view; note how the volume of the PVS increases its total percentage volume once the capillary has undergone rarefaction as in obesity, metabolic syndrome, and type 2 diabetes mellitus. (**D**) depicts the progression of a normal precapillary arteriole and postcapillary venule PVS to an EPVS once the capillary has undergone rarefaction, allowing for an increase in its total percentage volume of the PVS (1.–3.). (**B,C**) provided with permission from CC 4.0 [2]. ACef = perivascular astrocyte endfeet; AQP4 = aquaporin-4 (red bars); BEC = brain endothelial cells; BECact/dys = brain endothelial cell activation and dysfunction; CL =capillary lumen; EC = endothelial cell; lpsEVexos = lipopolysaccharide extracellular vesicle exosomes; NVU = neurovascular unit; Pcef = pericyte endfeet.

Notably, EPVS were defined as being enlarged by magnetic resonance imaging (MRI) when they measured 1–3 mm in diameter as a part of a position paper in 2013 that summarized the main outcomes of this international effort to provide the standards for reporting vascular changes on nEuroimaging (STRIVE) [48]. EPVS are thought to be associated with other radiological remodeling changes such as cerebral small vessel disease (SVD), which include lacunes and white matter hyperintensities (WMH) as well as microbleeds and microthrombi (Figure 7) [2,42,47].

| | EPVS | Lacunes | WMH |
|---|---|---|---|
| Location | Basal ganglia (BG) **Type I** Centrum semiovale (CSO) **Type II** Midbrain **Type III.** | Upper portions of Basal Ganglia thalamus, internal and external capsule, pons, and periventricular white matter. | Periventricular, deep white matter distinct from periventricular regions. |
| Morphology Shape | Well defined, round, oval, tubular. | Irregular shapes, sharp edges, Or wedged shaped. | Sharp edges, linear, and frequently follow the outlines of the adjacent ventricle. Elongated |
| Symmetry | Symmetrical | Asymmetrical | Asymmetrical |
| Size | 1-3mm | 3-15mm diameter | 3-12mm but may be larger; they are usually elongated |
| FLAIR (fluid-attenuated inversion recovery) | Primarily non-FLAIR | (+) FLAIR (+) FLAIR usually reflects siderosis or Gliosis - reactive astrocytes or both | (+) FLAIR |

**Figure 7.** Comparisons of enlarged perivascular spaces, lacunes, and white matter hyperintensities (WMH). mm = millimeter.

EPVS may develop abruptly or in a compensatory response, as occurs in sleep; however, when they occur with clinical disease they more commonly form in a sequence of events that is created over time in a spectrum of remodeling changes including SVD that is associated with small arteries, arterioles, capillaries, and venules of the brain with identifiable neuroimaging features on MRI including small subcortical infarcts or lacunes, WMH, and EPVS, which can be initially difficult to identify from one another (Figures 7 and 8) [2,4,42,47].

The concept of the PVU and the EPVS fit nicely with both the NVU concept and the emerging importance of the glymphatic system in regard to the clearance of neurotoxic metabolic waste.

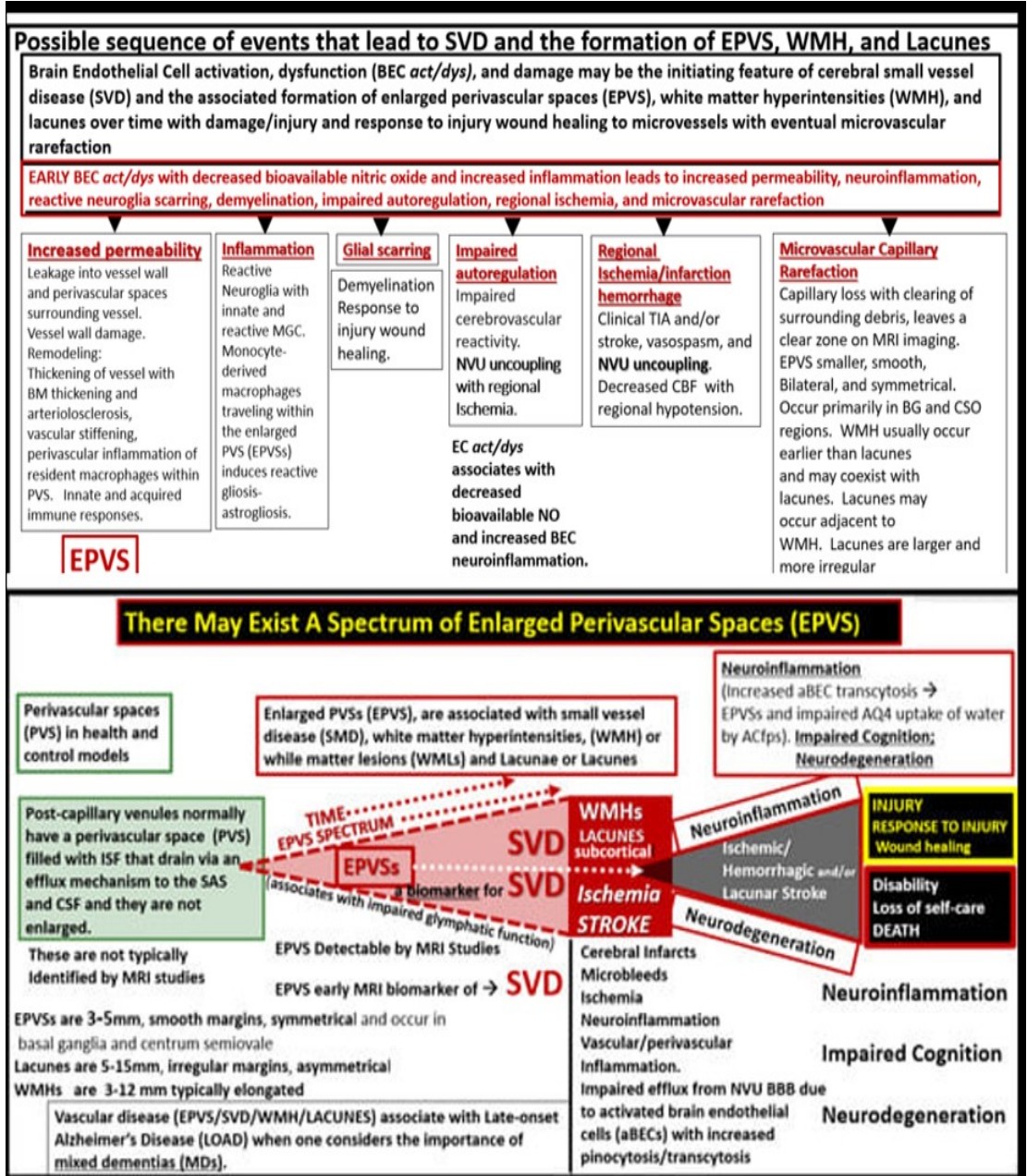

**Figure 8.** A possible sequence of events and an ongoing spectrum for the development of enlarged perivascular spaces (EPVS). Brain endothelial cell activation and dysfunction (BEC*act/dys*) and/or damage may be the initiating factor in this sequence of events, while attenuation and/or loss of the endothelial glycocalyx (ecGCx) may be concurrent or even precede BEC*act/dys*. Additionally, there exists a spectrum of remodeling changes and events in the development of EPVS that also includes lacunes, white matter hyperintensities (WMH), and cerebral small vessel disease. Images provided with permission from CC 4.0 [2]. aBECs = activated brain endothelial cells; ACfp = astrocyte endfeet; BBB = blood–brain barrier; BEC = brain endothelial cell; BG = basal ganglia; BM = basement membrane; CBF = cerebral blood flow; CSF = cerebrospinal fluid; CSO = centrum semiovale; ISF = interstitial fluid; LOAD = late-onset Alzheimer's disease; MGC = microglia cell; mm = micrometer; MRI = magnetic resonance imaging; NO = nitric oxide; NVU = neurovascular unit; PVS = perivascular spaces; SAS = subarachnoid space; TIA = transient ischemic attack; WMH = white matter hyperintensities.

#### 4. Protoplasmic Perivascular Astrocyte Endfeet (pvACef) Play a Crucial Role in the Development of the Perivascular Unit (PVU) and Enlarged Perivascular Spaces (EPVS)

As mentioned earlier, the pvACef are the master communicating cells via gap junction connexins Cx40, Cx43, and thus create a syncytium, connecting cells (see Figure 3), coupling cells as in NVU coupling and PVU coupling, and continuing and creating cells, so that they become continuous at the true capillary as it transitions to the postcapillary venule with normal PVS, and are creative in the development of the PVU and EPVS (Figure 9) [7–11,41].

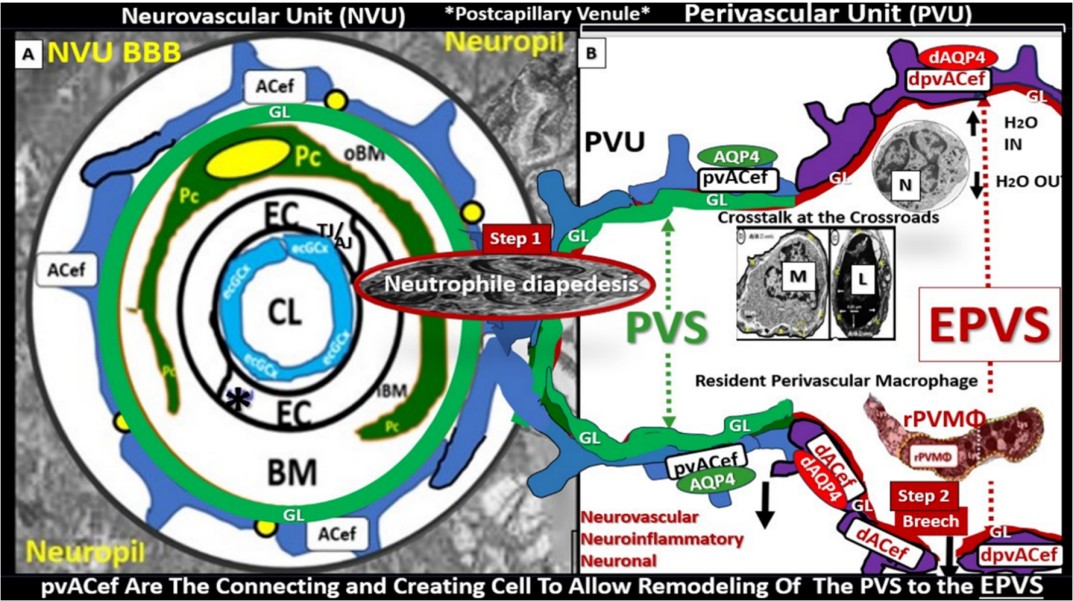

**Figure 9.** Comparison of the true capillary neurovascular unit (NVU) to the postcapillary venule perivascular unit (PVU). Perivascular astrocyte endfeet (pvACef) are the connecting and creating cells that allow remodeling of the perivascular units' (PVU) perivascular space (PVS) to transform into the enlarged perivascular space (EPVS). (**A**) illustrates the control neurovascular unit (NVU); note that when the BECs become activated and BBB disruption develops, this increases the permeability of fluids, peripheral cytokines and chemokines, and peripheral proinflammatory leukocytes with a neutrophile depicted herein penetrating the TJ/AJ paracellular spaces to enter the postcapillary venule along with monocytes and lymphocytes into the postcapillary PVU PVS. (**B**) initially depicts the normal PVU with its normal perivascular space (PVS) that undergoes remodeling transformation to become EPVS, allowing for close communicating crosstalk at the crossroad of the PVU. Note how the pvACef (pseudo-colored blue) and its glia limitans (pseudo-colored cyan and exaggerated in its thickness for illustrative purposes) face and adhere to the NVU BM extracellular matrix, and face the PVS PVU lumen since this has detached and separated and allowed the creation of a perivascular space that transforms to an EPVS. Also note how the glia limitans becomes pseudo-colored red once the EPVS have developed, and then allows the outer barrier of the PVS glia limitans to become breeched due to the activation of matrix metalloproteinases and the degradation of the proteins within the glia limitans, which allow neurotoxins and inflammatory cells to leak into the interstitial spaces of the neuropil and mix with the ISF and result in neuroinflammation, which (Step 2) of the two-step process of neuroinflammation. Yellow circles represent nanometer gaps between pvACef. Asterisk = tight and adherens junction; AP4 = aquaporin-4; BBB = blood–brain barrier; BM = both inner (i) and outer (o) basement membrane; dpvACef = dysfunctional astrocyte endfeet; EC = brain endothelial cell; ecGCx = endothelial glycocalyx; EVPS = enlarged perivascular space; GL = glia limitans; $H_2O$ = water; L = lymphocyte; M = monocyte; N = neutrophile; Pc = pericyte; PVS = perivascular space; PVU = perivascular unit; rPVMΦ = resident perivascular macrophage; TJ/AJ = tight and adherens junctions.

Owens et al. [49] have described a detailed two-step process of neuroinflammation, and this process hinges on the knowledge that fluids and solutes undergo diffusion and transfer primarily at the true capillary to the interstitium. Step 1 of the two-step process involves leukocyte transfer from the capillary lumen via primarily paracellular spaces to the perivascular unit of the postcapillary venules. Capillary leukocytes undergo primarily paracellular diapedesis into the PVU, where they remain until the signals (such as ROS-induced proteolysis) are able to degrade the outer barrier (the glia limitans) in order for the PVU PVS/EPVS leukocytes to breech the outer barrier and enter the ISS of the neuropil, Step 2 (Figures 9 and 10) [49,50].

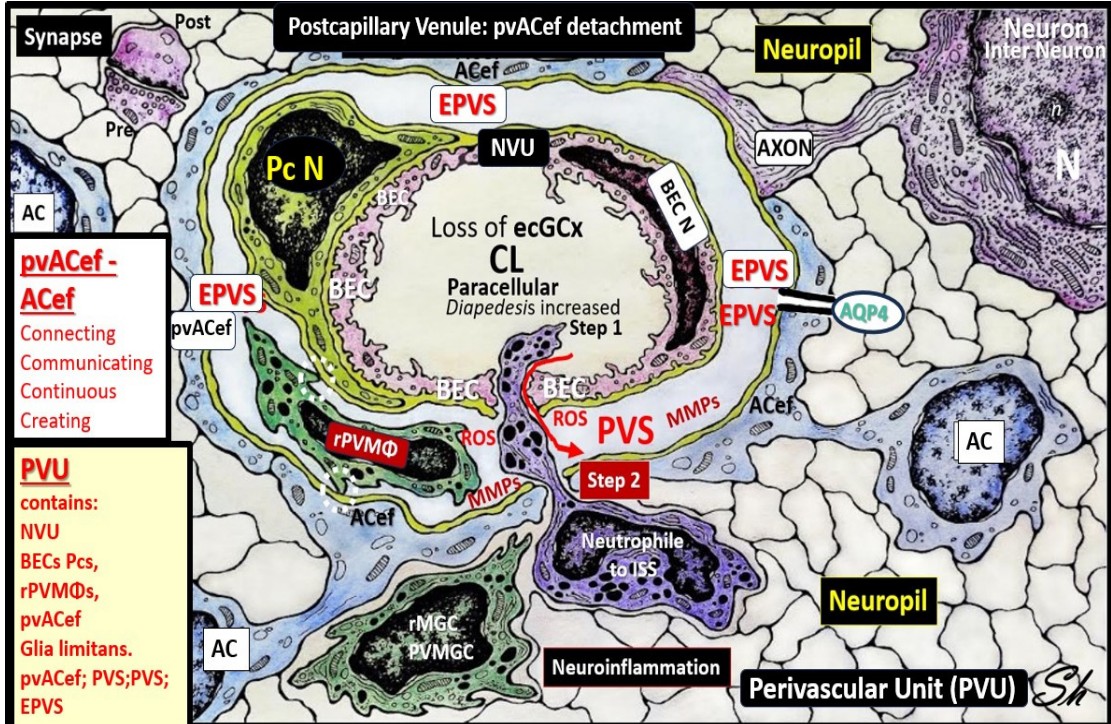

**Figure 10.** Perivascular astrocyte endfeet (ACef), neurovascular unit (NVU), perivascular unit (PVU), perivascular space PVS, and enlarged perivascular space (EPVS). The NVU is located centrally; note the absence of the endothelial glycocalyx (ecGCx) surface layer, which occurs in many neurovascular and neurodegenerative diseases with impaired cognition that also include obesity, metabolic syndrome (MetS), and type 2 diabetes mellitus (T2DM). Increased NVU permeability via BEC*act/dys* BBB disruption due to multiple clinical neurovascular and neurodegenerative diseases allows the entry of proinflammatory leukocytes into the PVU PVS in postcapillary venules. The accumulation of proinflammatory cells' oxidative stress with increased ROS will activate local and regional matrix metalloproteinase (MMP)—a proteolytic enzyme capable of degrading the glia limitans of the pvACef to allow the breeching of the postcapillary perivascular space and the entry of proinflammatory leukocytes, solutes, and neurotoxins into the interstitial spaces (ISSs) to result in neuroinflammation and increased *CNS*C/C, impaired cognition, and neurodegeneration via synaptic and neuronal loss with neural atrophy. Note the aquaporin water channel and also the isolated synapse (uncradled) in the upper left-hand side of illustration. This image is provided courtesy of CC by 4.0 [50]. AC = astrocyte; ACef = perivascular astrocyte endfeet; AQP4 = aquaporin-4; BEC = brain endothelial cell; N = nucleus; n = nucleolus; Pc = pericyte; PVU = perivascular unit; pvMGC; rMGC; rPVMΦ = resident perivascular macrophage; ROS = reactive oxygen species.

Additionally, Owens [49] has pointed out that the commonly observed histological phenomenon of perivascular cuffing in extensive neuroinflammation depicts the surrounding accumulation of proinflammatory leukocytes to reside within the perivascular spaces

by TEM. This finding further increases the crosstalk ability amongst these cells within the crossroads that Troili has discussed previously [41,49].

Lipopolysaccharide (LPS) injection is one of the best known and most commonly utilized methods to induce neuroinflammation [50–53]. Recently, Erickson et al. [54] utilized LPS to induce neuroinflammation in the brain, and found considerable transmission electron microscopy (TEM) remodeling changes in cortical layer III of the frontal regions at 7–10 weeks post treatment. These remodeling changes included BEC plasma membrane ruffling, increased extracellular microvesicles and small exosome formation, aberrant BEC mitochondria, and increased transcytosis with intact TJ/AJ; aberrant pericytes revealed Pc nucleus rounding and retracted cytoplasmic extensions, which attracted microglia cells to the NVU, and attenuated discontinuous endothelial glycocalyx, pvACef retraction and separation associated with development of EPVS (Figure 11) [50,54].

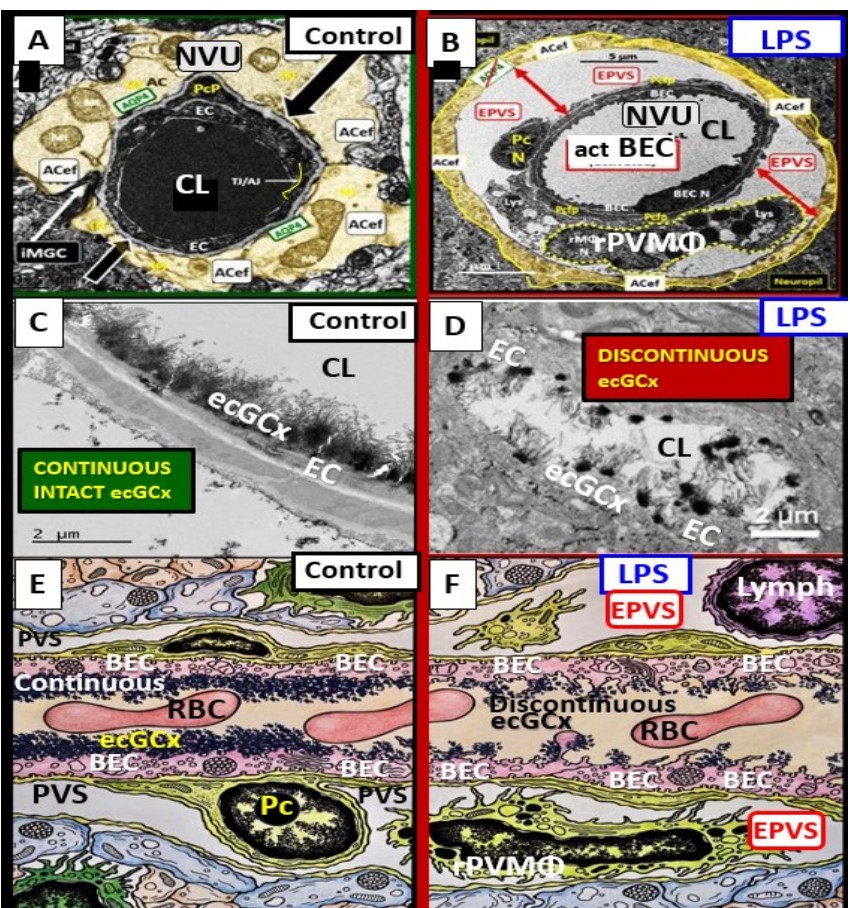

**Figure 11.** Examples of lipopolysaccharide (LPS)-induced neuroinflammation resulting in brain endothelial cell activation and dysfunction (BEC*act/dys*), loss of endothelial glycocalyx (ecGCx), and enlarged perivascular spaces (EPVS). (**A**,**C**,**E**) demonstrate the appearance control images. (**B**,**D**,**F**) depict LPS remodeling changes. (**B**) is an example of an EPVS (double red arrows) with a resident perivascular macrophage (rPVMΦ that appears reactive with multiple vesicles and vacuoles); also, note the detachment and retraction of the pseudo-colored yellow pvACef. (**D**) depicts the attenuation and/or loss of the ecGCx; note that the ecGCx is discontinuous as compared to the continuous ecGCx in control panel (**C**). (**F**) depicts the remodeling of the initial normal PVS to an EPVS in association with the aberrant discontinuous ecGCx. (**A**,**B**) are presented courtesy of CC 4.0 [3BEC] and (**C**,**D**) are presented courtesy of CC 4.0 [50]. ACEF = protoplasmic perivascular astrocyte endfeet; AQP4 = aquaporin-4; CL = capillary lumen; EC = brain endothelial cell; Lys = lysosome NVU = neurovascular unit; Pc = pericyte; Pcfp = pericyte foot processes; PVS = perivascular space; RBC = red blood cell; TJ/AJ = tight and adherens junction.

## 5. Perivascular Astrocyte Endfeet (pvACef)

For this section, one might refer to the NVU as the "neuro-glial-vascular unit" (NGVU), since the pvACs endfeet play such a critical role in connecting ACs to the NVU to accomplish NVU coupling with regional neurons to increase regional cerebral blood flow to neural activity [55,56]. Early on in our studies of the diabetic *db/db* mouse models at 20 weeks of age, our group found multiple ultrastructure remodeling changes including the reactive pvACef that were tightly adherent to the basement membrane in the control models and depicted ultrastructural detachment and retraction of the pvACef in the diabetic *db/db* models [7,57]. This detachment and retraction created a void electron lucent fluid-filled space around the NVU between the NVU BM and the pvACef glia limitans (Figure 12) [7,57].

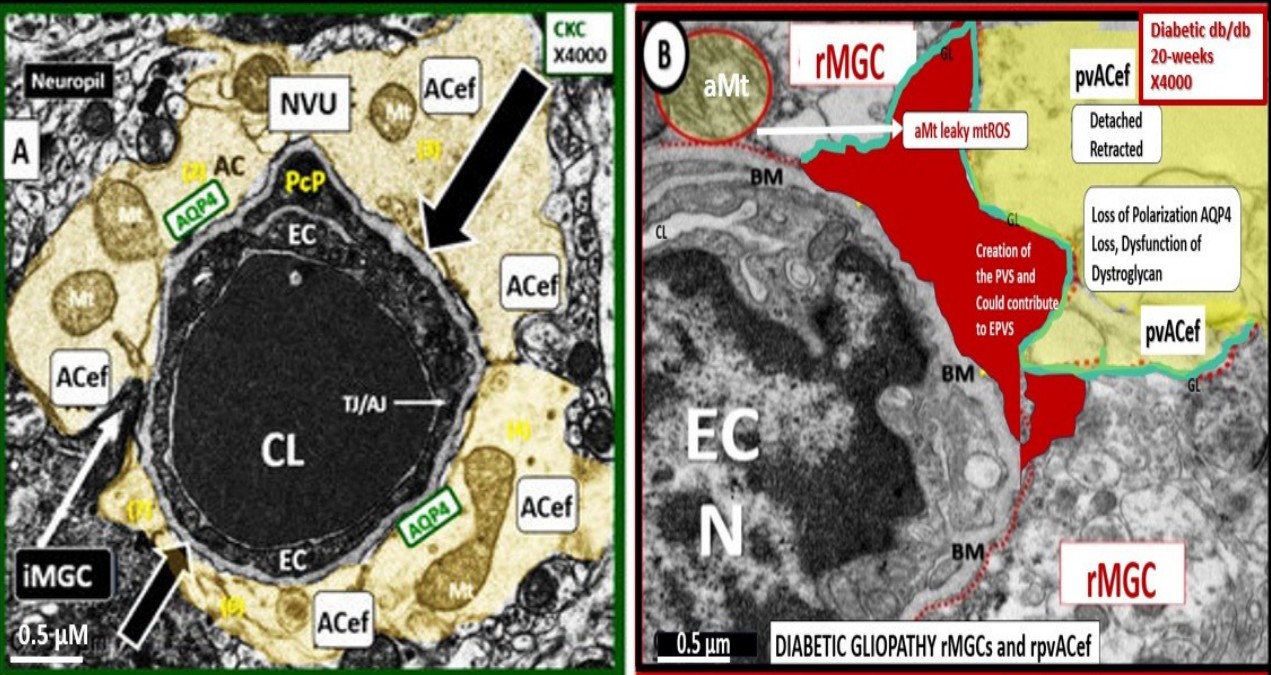

**Figure 12.** Detachment and retraction of the reactive perivascular astrocyte endfeet (rpvACef) from the neurovascular unit (NVU) of the 20-week-old diabetic *db/db* models. (**A**) demonstrates the normal control appearance of the NVU with its tightly adherent pvACef that contain the polarized extracellular matrix basement membrane receptor proteins beta-dystroglycan and integrins of the surrounding pvACef at the plasma membrane to maintain this normal anatomical ultrastructure. Also note the aquaporin-4 that is polarized along with beta-dystroglycan and integrins at the plasma membrane of the pvACef. (**B**) depicts the detachment and retraction of the reactive pvACef (rpvACef) in diabetic female *db/db* models at 20 weeks of age from the frontal regions in layer III. Note the adjacent reactive microglia cell(s) (rMGC) containing an aberrant mitochondrion (aMt). Importantly, note the red pseudo-colored space that is created upon the rpvACef detachment and retraction. Also, note that the detached retracted pvACef has been pseudo-colored yellow to match the pvACef in panel A. This type of rpvACef remodeling was a common finding in the diabetic *db/db* models at 20 weeks of age. Modified (**A**,**B**) were provided with permission from CC 4.0 [7]. The faint yellow numbers are intended to number the pvACef (1-7). Scale bar = 0.5μm. AC = astrocyte; ACef = protoplasmic perivascular astrocyte endfeet; aMt = aberrant mitochondria; AQP4 = aquaporin-4 water channel; BM = basement membrane; CKC = control model; CL = capillary lumen; EC = brain endothelial cell; iMGC = interrogating microglia cell; N = nucleus; NVU = neurovascular unit; PcP = pericyte processes; pvACef = protoplasmic perivascular astrocyte endfeet; rMGC = reactive microglia cell; rpvACef = reactive perivascular endfeet; TJ/AJ = tight and adherens junctions.

This detachment and retraction are currently felt to be a result of the degradation and/or loss of function of the extracellular matrix receptors beta-dystroglycan (β-DG) and integrin alpha 6 beta 4 (α6β4) proteins localized to the plasma membrane of the pvACef due to oxidative stress via ROS that induce the proteolytic matrix metalloproteinases (MMP-2, 9) [58–63]. Importantly, the β-DG and α6β4 integrin receptors of the pvACef secure it to the BM via its connections that adhere to the laminin and other cytoskeletal components of the ECM BM of the NVU [58], and the α-dystroglycan form is responsible for the linkage to the basement membrane proteins [61], whereas β-dystroglycan links α-dystroglycan to the actin cytoskeleton [63]. Also, DG proteins are known to be present on dendritic spines [62].

It is a fascinating perspective that among the billions of neuroglia and neurons, the mammalian brain has interlaced an elaborate network of blood vessels that are enwrapped specifically by pvACef and connected to the neuronal synapses by psACef processes (80–90%) to provide a plentiful blood supply.

## 6. Perisynaptic Astrocyte Endfeet (psACef), Aquaporin-4 (AQP4), Impaired Synaptic Transmission and Synaptopathy

Given that astroglia are the most abundant and voluminous cells in the CNS, and that they project very long processes to both the vasculature and neurons, it is appropriate to better understand how they are capable of signaling and regulating distal vascular cells, synapses of neurons and their functions, and especially how they protect and support them [8,9]. Currently, it is commonly accepted that psACef are both structural and functional components of synapses throughout the brain [9].

Detachment of the cradling perisynaptic astrocyte endfeet may occur similar to the detachment of the pvACef due to the loss of function of aquaporin-4 (AQP4) by either ROS-induced MMP degradation associated with neuroinflammation or loss of plasma membrane AQP4 polarity due to neuroinflammation and/or BBB disruption in addition to the loss of function of the dystroglycans as discussed above in regards to the pvACef (Figure 13) [8,9,11,64,65].

Normal healthy control model psACef are known to take up excess generated water/fluid during synaptic activity in cradled neurons, and if AQP4 is dysfunctional due to loss of polarization, water/fluid may accumulate to allow an increased expansion and support further detachment of the psACef to increase the distance between the synaptic cleft and the psACef and support further separation and dysfunction of the psACef [8,9]. Thus, any disruption or dysfunction of the synaptic cradle is capable of resulting in a synaptopathy that could contribute to impaired synaptic transmission and impaired cognition with neurodegeneration [9].

AQP4 water channels localized to the plasma membrane of the pvACef and psACef play an important role in controlling fluid homeostasis within the ECM between cells and structures such as the NVU and the synaptic cradle in psACef in the CNS [9,65]. They also play critical roles in the function of the glymphatic system [17] and provide homeostasis for the entire CSF network as in the whole brain (Figures 1 and 3).

While ACs play numerous important roles in the brain as discussed in the introduction and throughout this review, they are critically important for controlling the volume of CNS, CSF, ISF, PVS, glymphatic space for waste removal, as well as their own size and volume, due to their highly polarized plasma membrane bidirectional water channel AQP4 [44]. AQP4's role in maintaining normal CNS homeostasis includes potassium buffering, cerebrospinal fluid circulation, interstitial fluid resorption, regulation of extracellular space volume, waste clearance, neuroinflammation, osmosensing, $Ca^{2+}$ signaling, and cell migration, and it is also required for the normal function of the retina, inner ear, and olfactory system (Figure 14) [8,66].

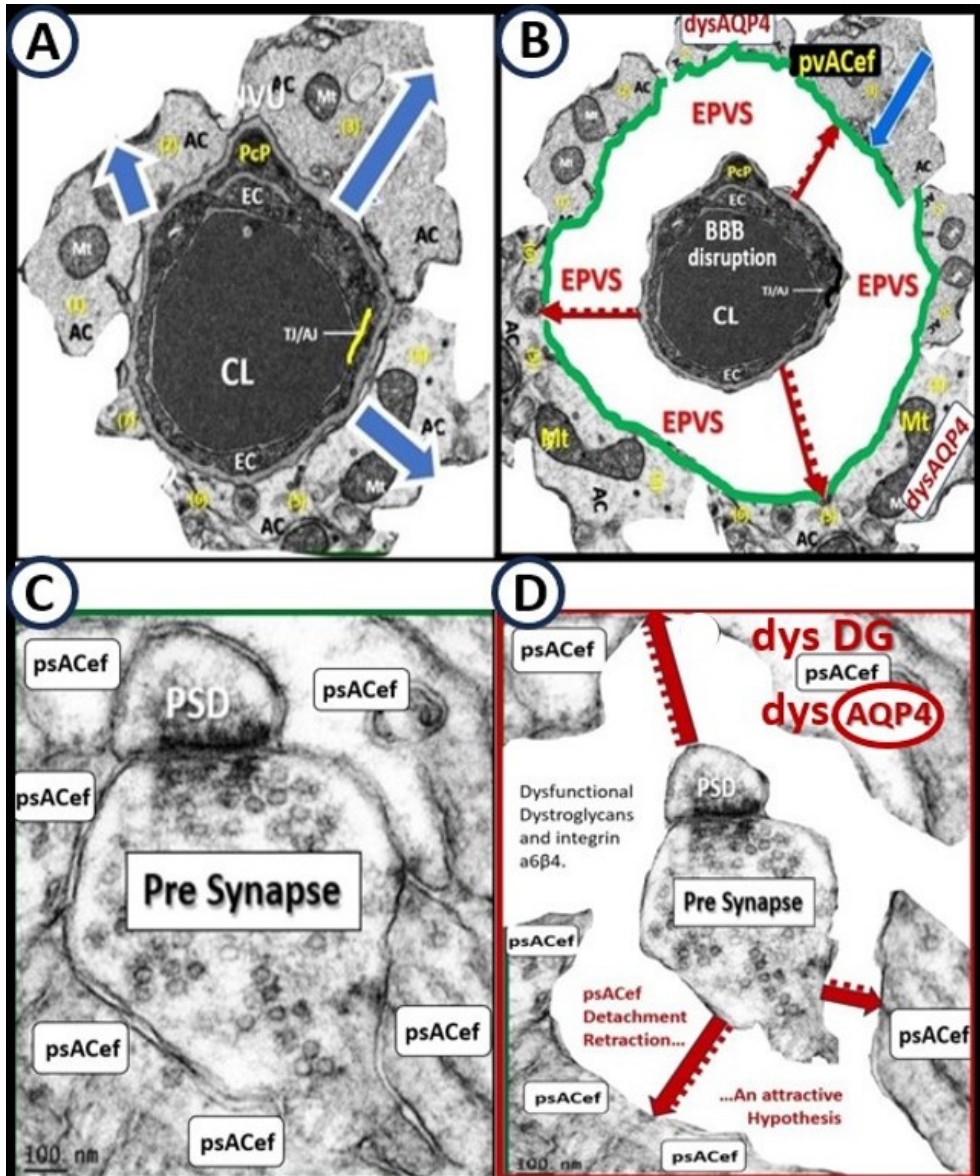

**Figure 13.** Similarities and comparisons between perivascular astrocyte endfeet (pvACef) and the cradling perisynaptic astrocyte endfeet (psACef) detachment and separation. These similarities implicate damaged or dysfunction aquaporin-4 (AQP4) either due to activated proteases such as matrix metalloproteinases (MMP-2, 9) or to a loss of polarization of AQP4 from the plasma membranes resulting in impaired synaptic transmission and impaired cognition or the timing of arrival of incoming information to disturb networks of information. (**A**,**C**) are female age-matched controls and (**B**,**D**) are from 20-week-old diabetic *db/db* models with tissues obtained from the frontal cortex, cortical layer III, and depict detachment and separation of pvACef in (**B**) and psACef in panel D. Note that this detachment and separation create a perivascular space (PVS) (**B**) and a perisynaptic space (**D**) that may continue to become enlarged with dysfunctional dystroglycan (dysDG) and dysfunctional aquaporin-4 (dysAQP4) as in (**panels B**,**D**). Note that the cyan green line denoting the glia limitans in (panel **B**) is not present in (**D**). Blue arrows represent detachment and separation of pvACef. Red dotted arrows represent detachment and separation with emphasis. Faint yellow numbers indicate numbering of pvACef (1-7). Images in A and B are reproduced courtesy of CC 4.0 [7]. Scale bars =100 nm (**D**,**E**). BBB = blood–brain barrier; CL = capillary lumen; dys = dysfunctional; DG = dystroglycans; EC = brain endothelial cell; NVU = neurovascular unit; PcP = pericyte process; PSD = post synaptic density; TJ/AJ = tight and adherens junctions.

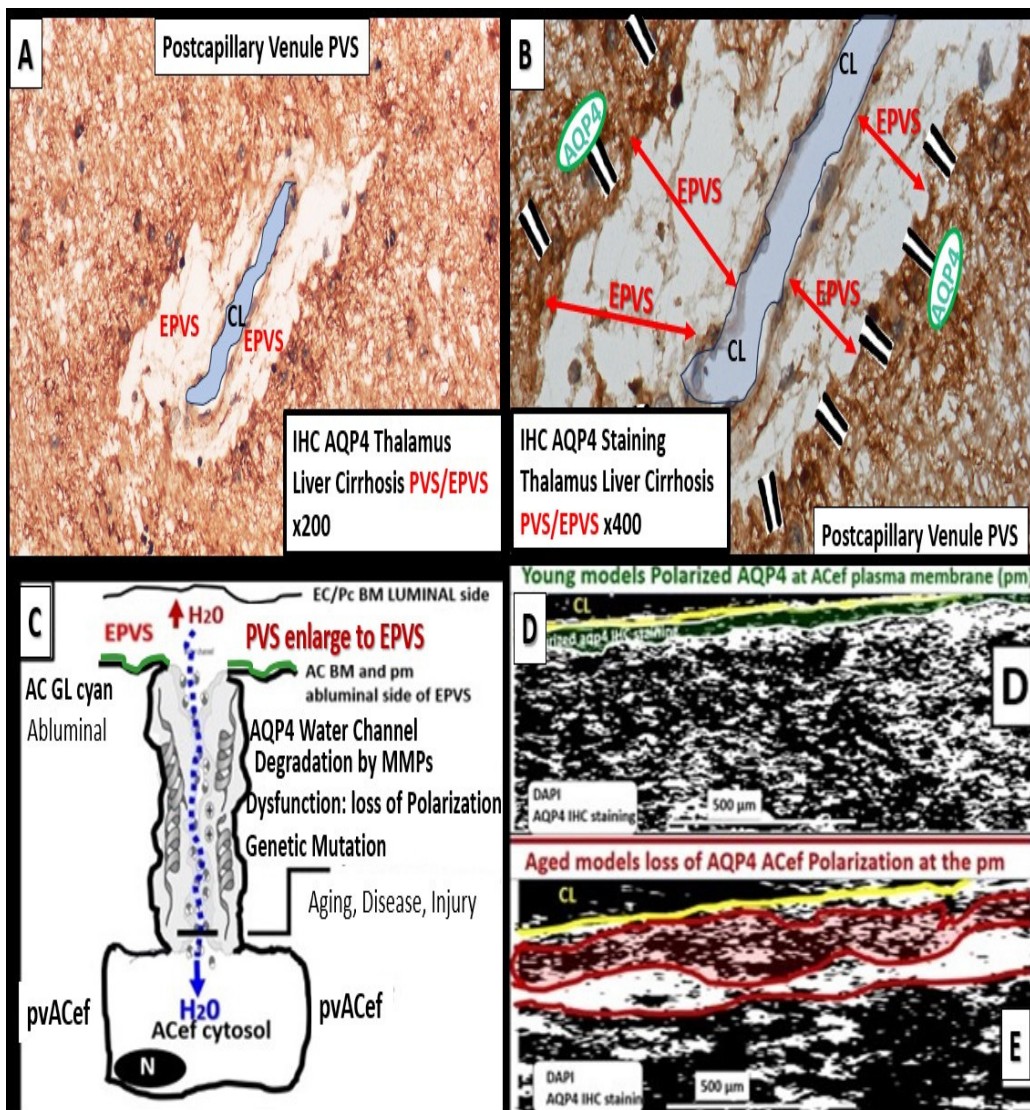

**Figure 14.** The perivascular astrocyte endfeet (pvACef) with their polarized aquaporin-4 (AQP4) water channels delimit the abluminal perivascular unit (PVU) with its perivascular spaces/enlarged perivascular spaces (PVS/EPVS) and perisynaptic astrocyte endfeet (psACef). (**A,B**) demonstrate by immunohistochemical staining the presence of AQO4 in the pvACef. (**C**) is a schematic of the AQP4 channel and illustrates water moving into the PVS to contribute to the PVS enlargement. (**D**) illustrates in younger models that AQP4 is tightly polarized to the plasma membrane of the pvACef as compared to (**E**), which depicts a loss of AQP4 polarization in older models. Modified image provided with permission from CC 4.0 [44]. Scale bar = 500. CL = capillary lumen; IHC = immunohistochemistry; N = nucleus.

Deficient AQP4 has been shown to result in impaired synaptic plasticity and transmission [67–70]. Clinically, there are two neurological diseases that are associated with antibodies against AQP4 water channels: neuromyelitis optica [71,72] and neuromyelitis optica spectrum disorder [73,74].

Dysfunction or loss of polarization and function of AQP4 water channels or genetic knock-out models of AQP4 result in glymphatic system dysfunction via brain-wide interstitial fluid stagnation [66,74–76]. Nielsen et al. were able to demonstrate that nanogold particle staining of AQP4 by transmission electron immunochemistry was localized to the plasma membrane of the pvACef where they tightly adhered to the NVU basement membrane [77].

## 7. Impaired Perivascular Astrocyte Endfeet (pvACef), AQP4, Glymphatic System (GS), and Clearance of Metabolic Waste (MW) Associated with Cerebral Small Vessel Disease (SVD)

Dysfunctional pvACef associated with impaired AQP4 water channels, EPVS, and glymphatic system impairment with impaired MW clearance are involved in the development of SVD [78]. Importantly, glymphatic function has now been found to be impaired in many processes relevant to SVD, which include aging [79], microinfarctions [80], stroke [81], late-onset Alzheimer's disease [82], migraine headaches [83], and diabetes [34]. There are estimated to be 46.8 million people who suffer from dementia globally, and SVD is known to be present in each of them [78,84]. Since SVD can be detected as a marker by MRI for future dementia and often precedes the clinical manifestations of transient ischemic attacks and strokes by years or decades, it is worthwhile to better understand the role of these multi-factorial dysfunctions and their mechanisms as discussed in this review as a potentially preventable cause of cerebrovascular disease and dementia.

There are three basic types of SVD, namely Type **1**: sporadic arteriolosclerosis that is associated with aging and systemic hypertension (especially systolic hypertension and T2DM); Type 2: sporadic hereditary cerebral amyloid angiopathy (CAA); and Type 3: inherited or genetic forms (not CAA), most commonly cerebral autosomal dominant arteriopathy with subcortical ischemic strokes and leukoencephalopathy (CADASIL) [85]. Additionally, EPVS observed on brain MRIs are thought to be a marker of glymphatic dysfunction and reflect impairment of brain fluid and waste clearance [86]. Further, cerebral capillary rarefaction (Figure 6) has been determined to precede WMH in genetic models of cerebral ischemic SVD [87]; also, a genomics study of EPVS supported early mechanisms of SVD [88].

Indeed, the new perspective of the GS with impaired clearance of MW provides an additional mechanism regarding the pathophysiology of SVD and its associated clinical diseases including mixed dementias [21,78,89].

## 8. Conclusions

EPVS are strongly related to aging. We are currently one the oldest populations in history, and this will persist for at least the next two decades [2,21,23,50]. Notably, it has been estimated that approximately two billion individuals of the global population will exceed 60 years of age by 2050, which is 1/5 of the world's population [89]. Therefore, it is essential that we strive to learn as much as possible regarding the development of EPVS as they transition from normal PVS in both the precapillary perivascular arteries, arterioles, true capillaries without a PVS, and the postcapillary venules, which serve as the anatomical conduit for the glymphatic system to remove neurotoxic metabolic waste from neuronal metabolic activity via the PVS glymphatic space. As we gain a better understanding of the possible mechanisms as to how the normal PVS transitions to pathological EPVS in regards to the pvACef and psACef, we may be able to slow or prevent the development of EPVS as well as the associated neurovascular and neurodegenerative diseases that are so devastating to the welfare of our aging global populus.

**Funding:** This research received no external funding.

**Institutional Review Board Statement:** The tissues provided for the representative electron microscopic images utilized in this manuscript were all approved in advance by the University of Missouri Institutional Animal Care and Use Committee (No. 190). The animals were cared for in accordance with National Institutes of Health guidelines and by the Institutional Animal Care and Use Committees at the Harry S. Truman Memorial Veterans' Hospital and the University of Missouri, Columbia, MO, USA, which conformed to the Guide for the Care and Use of Laboratory Animals published by the National Institutes of Health (NIH).

**Informed Consent Statement:** Not applicable.

**Data Availability Statement:** The data and materials can be provided upon reasonable request.

**Acknowledgments:** The author would like to acknowledge Tatyana Shulyatnikova for the contribution of many artistic illustrations and her editing of this manuscript. The author would also like to acknowledge the DeAna Grant Research Specialist of the Electron Microscopy Core Facility at the Roy Blunt NextGen Precision Health Research Center, University of Missouri, Columbia, Missouri. The author also acknowledges the kind support of the William A. Banks Lab at the VA Medical Center, Seattle, Washington.

**Conflicts of Interest:** The author declares no conflict of interest.

**Abbreviations**

AC: astrocyte; ACef, astrocyte endfeet; AGE/RAGE, advanced glycation end products/receptor for advanced glycation end products; AQP4, aquaporin-4; BBB, blood–brain barrier; BEC(s), brain endothelial cell(s); BECact/dys, brain endothelial cell activation/dysfunction; BM, basement membrane; CL, capillary lumen; EPVS, enlarged perivascular spaces; GSH, glutathione; GS, glymphatic space; ISF, interstitial fluid; ISS, interstitial space; LOAD, late-onset Alzheimer's disease; LPS, lipopolysaccharide; MetS, metabolic syndrome; MGCs, microglia cells; MMP-2,-9, matrix metalloproteinase-2,-9; MW, metabolic waste; NVU, neurovascular unit; Pc, pericyte; Pcfp, pericyte foot process; PVS, perivascular spaces; PVS/EPVS, perivascular space/enlarged perivascular space; RSI, reactive species interactome; rPVMΦ, resident perivascular macrophages; SAS, subarachnoid space; SOD, super oxide dismutase; SVD, cerebral small vessel disease rPVMΦ, reactive perivascular macrophage;; T2DM, type 2 diabetes mellitus; TEM, transmission electron microscopy; TGFβ, transforming growth factor beta; TI/AJs, tight and adherens junctions; VAD, vascular artery disease; WMH, white matter hyperintensities.

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
