# Peer review of "Protoplasmic Perivascular Astrocytes Play a Crucial Role in the Development of Enlarged Perivascular Spaces in Obesity, Metabolic Syndrome, and Type 2 Diabetes Mellitus"

_2571-6980, doi:10.3390/neuroglia4040021_

Round 1

Reviewer 1 Report

Comments and Suggestions for Authors

Although the review concerning “Protoplasmic perivascular astrocytes play a crucial role in the development of enlarged perivascular spaces in obesity, metabolic syndrome, and Type 2 diabetes mellitus” was very interesting, numbers of points need clarifying and certain statements require further justification. These are given below.

<Point>

1.      In Figure 2, scale bar(s) should be added. Figures are usually magnified or reduced by printer. Therefore, scale bar(s) should be added in the figure(s).

2.      In Figure 3, scale bar(s) should be added. Figures are usually magnified or reduced by printer. Therefore, scale bar(s) should be added in the figure(s).

3.      “15, 16]” (line 171) should be reduced and the underline should be deleted.

4.      Figure numbers in page 6, 7, 8, 9, 10, 11, 12, 14, 15, 17, and 19 should be “Figure 4, 5, 6, 7, 8, 9, 10, 11, 12, 13, and 14”, respectively.

5.      “Funding” should be described.

6.      In Ref. 26, “American Journal of Alzheimer’s Disease & Other Dementias. 2020;35” should be changed to “Am J Alzheimers Dis Other Demen 2020;35:1533317520912126”.

7.      In Ref. 72, “Jarius S, Paul F, Weinshenker BG, Levy M, Kim HJ, Wildemann B. Neuromyelitis optica” should be changed to “Jarius S, Paul F, Weinshenker BG, Levy M, Kim HJ, Wildemann B. Neuromyelitis optica. Nat Rev Dis Primers 2020;6:85”.

8.      Ref. 73 “Nat Rev Dis Primers. 2020;6(1):85. doi: 10.1038/s41572-020-0214-9” should be deleted. Refs. 74-92 should be changed to 73-91.

Author Response

Response to reviewer number 1

Manuscript ID: neuroglia-2664473

First, author wishes to thank reviewer number one for the time, effort and knowledge required to review this submitted manuscript.

The authors response to reviewer recommendations will be in blue coloring in the revised submitted manuscript and in this response to reviewer.

Although the review concerning “Protoplasmic perivascular astrocytes play a crucial role in the development of enlarged perivascular spaces in obesity, metabolic syndrome, and Type 2 diabetes mellitus” was very interesting, numbers of points need clarifying and certain statements require further justification. These are given below.

<Point>

  1. In Figure 2, scale bar(s) should be added. Figures are usually magnified or reduced by printer. Therefore, scale bar(s) should be added in the figure(s). Author has rebuilt Figure 2 and the scale bars are now readily evident in the new figure 2 in the revised manuscript
  2. In Figure 3, scale bar(s) should be added. Figures are usually magnified or reduced by printer. Therefore, scale bar(s) should be added in the figure(s). Author has rebuilt Figure 3 and the scale bars are now readily evident in the new figure 3 in the revised manuscript.
  3. “15, 16]” (line 171) should be reduced and the underline should be deleted. Author has made the necessary corrections In the revised manuscript.
  4. Figure numbers in page 6, 7, 8, 9, 10, 11, 12, 14, 15, 17, and 19 should be “Figure 4, 5, 6, 7, 8, 9, 10, 11, 12, 13, and 14”, respectively. Author has corrected these errors and oversights in the submitted revised manuscript.
  5. “Funding”should be described. This has been corrected as follows:  Author has made the following corrections that read as follows:  : Author received no funding.
  6. In Ref. 26, “American Journal of Alzheimer’s Disease & Other Dementias. 2020;35” should be changed to “Am J Alzheimers Dis Other Demen 2020;35:1533317520912126”. Author has made the required changes as follows in reference 26. As follows: 26. Wu D, Yang X, Zhong P, Ye X, Li C, Liu X. Insulin Resistance Is Independently Associated With Enlarged Perivascular Space in the Basal Ganglia in Nondiabetic Healthy Elderly Population. Am J Alzheimers Dis Other Demen. 2020:35:1533317520912126. doi: 10.1177/1533317520912126 

  1. In Ref. 72, “Jarius S, Paul F, Weinshenker BG, Levy M, Kim HJ, Wildemann B. Neuromyelitis optica” should be changed to “Jarius S, Paul F, Weinshenker BG, Levy M, Kim HJ, Wildemann B. Neuromyelitis optica. Nat Rev Dis Primers 2020;6:85”. Author has made the following corrections:  Jarius S, Paul F, Weinshenker BG, Levy M, Kim HJ, Wildemann B. Neuromyelitis optica. Nat Rev Dis Primers. 2020;6(1):85. doi: 10.1038/s41572-020-0214-9

  1. Ref. 73 “Nat Rev Dis Primers. 2020;6(1):85. doi: 10.1038/s41572-020-0214-9” should be deleted. Refs. 74-92 should be changed to 73-91. Author has made the following changes as follows: Mader S, Brimberg L. Aquaporin-4 Water Channel in the Brain and Its Implication for Health and Disease. Cells. 2019;8(2): 90. doi: 10.3390/cells8020090

The references have been renumbered and matched to the text.

Author is very appreciative of the time, effort, and knowledge required to edit this submitted manuscript .

Sincerely with gratitude,

Melvin R Hayden

Submitting author

Reviewer 2 Report

Comments and Suggestions for Authors

The manuscript submitted by Hayden addresses a rather uncommon neurobiological theme the role of perivascular astrocytes in the development of Perivascular Spaces in metabolic pathologies.  It is a well written review that needs only minor corrections such the elimination of the adjective human when referring to the fact that astrocytes are the most abundant brain cells in vertebrates (abstract). 

Comments on the Quality of English Language

The usage of English in this manuscript is correct.

Author Response

Response to reviewer number 2

Manuscript ID: neuroglia-2664473

First, author wishes to thank reviewer number two for the time, effort and knowledge required to review this submitted manuscript.

The authors response to reviewer recommendations will be in blue coloring in the revised submitted manuscript and in this response to reviewer.

The manuscript submitted by Hayden addresses a rather uncommon neurobiological theme the role of perivascular astrocytes in the development of Perivascular Spaces in metabolic pathologies.  It is a well written review that needs only minor corrections such the elimination of the adjective human when referring to the fact that astrocytes are the most abundant brain cells in vertebrates (abstract).  Author has made the following corrections in the first sentence of the abstract as follows:  Abstract:  Astrocytes (ACs) are the most abundant cell in the brain and importantly, are the master connecting and communicating cells that provide structural and functional support of brain cells at all levels of organization.

Author is very appreciative of the time, effort, and knowledge required to edit this submitted manuscript.

Sincerely with gratitude,

Melvin R Hayden

Submitting author

Response to reviewer number 2

Manuscript ID: neuroglia-2664473

First, author wishes to thank reviewer number two for the time, effort and knowledge required to review this submitted manuscript.

The authors response to reviewer recommendations will be in blue coloring in the revised submitted manuscript and in this response to reviewer.

The manuscript submitted by Hayden addresses a rather uncommon neurobiological theme the role of perivascular astrocytes in the development of Perivascular Spaces in metabolic pathologies.  It is a well written review that needs only minor corrections such the elimination of the adjective human when referring to the fact that astrocytes are the most abundant brain cells in vertebrates (abstract).  Author has made the following corrections in the first sentence of the abstract as follows:  Abstract:  Astrocytes (ACs) are the most abundant cell in the brain and importantly, are the master connecting and communicating cells that provide structural and functional support of brain cells at all levels of organization.

Author is very appreciative of the time, effort, and knowledge required to edit this submitted manuscript.

Sincerely with gratitude,

Melvin R Hayden

Submitting author

Response to reviewer number 2

Manuscript ID: neuroglia-2664473

First, author wishes to thank reviewer number two for the time, effort and knowledge required to review this submitted manuscript.

The authors response to reviewer recommendations will be in blue coloring in the revised submitted manuscript and in this response to reviewer.

The manuscript submitted by Hayden addresses a rather uncommon neurobiological theme the role of perivascular astrocytes in the development of Perivascular Spaces in metabolic pathologies.  It is a well written review that needs only minor corrections such the elimination of the adjective human when referring to the fact that astrocytes are the most abundant brain cells in vertebrates (abstract).  Author has made the following corrections in the first sentence of the abstract as follows:  Abstract:  Astrocytes (ACs) are the most abundant cell in the brain and importantly, are the master connecting and communicating cells that provide structural and functional support of brain cells at all levels of organization.

Author is very appreciative of the time, effort, and knowledge required to edit this submitted manuscript.

Sincerely with gratitude,

Melvin R Hayden

Submitting author

Response to reviewer number 2

Manuscript ID: neuroglia-2664473

First, author wishes to thank reviewer number two for the time, effort and knowledge required to review this submitted manuscript.

The authors response to reviewer recommendations will be in blue coloring in the revised submitted manuscript and in this response to reviewer.

The manuscript submitted by Hayden addresses a rather uncommon neurobiological theme the role of perivascular astrocytes in the development of Perivascular Spaces in metabolic pathologies.  It is a well written review that needs only minor corrections such the elimination of the adjective human when referring to the fact that astrocytes are the most abundant brain cells in vertebrates (abstract).  Author has made the following corrections in the first sentence of the abstract as follows:  Abstract:  Astrocytes (ACs) are the most abundant cell in the brain and importantly, are the master connecting and communicating cells that provide structural and functional support of brain cells at all levels of organization.

Author is very appreciative of the time, effort, and knowledge required to edit this submitted manuscript.

Sincerely with gratitude,

Melvin R Hayden

Submitting author
